# Life-Cycle Assessment of LEED-CI v4 Projects in Shanghai, China: A Case Study

Svetlana Pushkar

Department of Civil Engineering, Ariel University, Ariel 40700, Israel; svetlanap@ariel.ac.il

**Abstract:** The purpose of this study was to explore green office building certification strategies in Shanghai. The 45 LEED-CI v4 gold-certified office space projects were sorted by energy and atmosphere credit (EAc6, optimize energy performance) into two groups: 15 projects with the lowest EAc6 achievement (Group 1) and 15 projects with the highest EAc6 achievement (Group 2). To reach the gold certification level in Group 1, high achievement in EAc6 is associated with low achievement in two indoor environmental quality credits (EQc2, low-emitting materials, and EQc8, quality views), while in Group 2, low achievement in EAc6 is associated with high achievement in EQc2 and EQ8. For the life-cycle assessment (LCA), the functional unit was designated as follows: production (P) stage: production of building materials needed to ensure the requirements of EQc2 and EQc8 for 1 m$^2$ of the building area; and operational energy (OE) stage: OE of 1 m$^2$ of the building area over 50 years of the building's lifetime. For the OE stage, two fuel source scenarios were used: 71.07% coal + 28.08% natural gas + 0.81% wind power (WP) + 0.04% photovoltaic (PV) (Scenario 1) and 50% WP + 50% PV (Scenario 2). The results of the LCA (P + OE) showed that under Scenario 1, the LEED certification strategy in Group 1 was greener than that in Group 2. When using Scenario 2, no differences were found between the two groups.

**Keywords:** certification strategies; life-cycle assessment; energy and atmosphere credits; indoor environmental quality credits; ReCiPe2016





## 1. Introduction

### 1.1. Description of the Problem

In 2002, Trusty and Horst [1] addressed the problem of integrating life-cycle assessment (LCA) into the Leadership in Energy and Environmental Design (LEED) green rating system. They used an 18-story office tower with five levels of underground parking, designed as a conventional reinforced concrete structure with a curtain wall exterior cladding system, to measure six environmental impacts: embodied energy, solid waste, air pollution, water pollution, global warming potential, and weighted resource use. They noted, "If the design team comes up with a building design (in which each environmental impact) is halved, they won't get a single LEED score". In 2016, De Wolf et al. [2] noted that buildings certified as platinum by LEED (the highest level of certification, ≥80 points) had the highest material usage and the highest environmental impact, while buildings with the lowest certification level (40–49 points) had the lowest material usage and impact. This case study illustrates that LEED does not currently reward buildings with lower embodied impact.

Table 1 shows the abbreviations and full names of the categories and credits for the LEED for Commercial Interiors version 4 (LEED-CI v4) green rating system and the terminology for LCA used in this study.

**Table 1.** LEED-CI v4 and LCA terminology.

| Abbreviation | LEED Category/Credit or LCA Terminology | LEED/LCA |
|---|---|---|
| IP | Integrative process | LEED category |
| LT | Location and transportation | LEED category |
| WE | Water efficiency | LEED category |
| EA | Energy and atmosphere | LEED category |
| MR | Materials and resources | LEED category |
| EQ | Indoor environmental quality | LEED category |
| IO | Innovation | LEED category |
| RP | Regional priority | LEED category |
| LTc3 | Access to quality transit | LT credit |
| EAc4 | Enhanced refrigerant management | EA credit |
| EAc6 | Optimize energy performance | EA credit |
| MRc2 | Interiors life-cycle impact reduction | MR credit |
| MRc5 | Building product disclosure and optimization, material ingredients | MR credit |
| EQc2 | Low-emitting materials | EQ credit |
| EQc8 | Quality views | EQ credit |
| FU | Functional unit | LCA term |
| P | Production stage | LCA term |
| OE | Operational energy | LCA term |
| LCI | Life cycle inventory | LCA |
| LCIA | Life cycle impact assessment | LCA |
| ReCiPe2016 | LCIA method | LCA |
| WP | Wind power | Fuel source |
| PV | Photovoltaic | Fuel source |

*1.2. LEED Version 4 Green Rating System*

The LEED v4 green rating system was launched in the US in 2013 [3]. The LEED v4 system contains 39 sub-systems, including LEED-NC (new construction), LEED-EB (existing building), LEED-C and -S (core and shell), LEED-CI, and so on. In this study, the author focused on LEED-CI v4 gold-certified office-space projects, since this type of project is widespread in both the US and in China [4]. The LEED-CI v4 sub-system contains six main categories and two bonus categories [5]. The main categories are integrative process (IP), with a maximum of 2 points; location and transportation (LT), with a maximum of 18 points; water efficiency (WE), with a maximum of 12 points; energy and atmosphere (EA), with a maximum of 38 points; materials and resources (MR), with a maximum of 13 points; and indoor environmental quality (EQ), with a maximum of 17 points. The two bonus categories are innovation (IO), with a maximum of six points, and regional priority (RP), with a maximum of four points. Each category contains one or more credits.

*1.3. Relationship between LEED Certification and LCA Results*

An analysis of the literature shows that there are at least two types of relationships between LCA and LEED-certified projects. The first type is the incorporation of LCA into the LEED rating system (LCA–LEED model) [6]. The second type is the conversion of LEED scores to LCA to identify differences in the strategies of LEED-certified projects [7,8].

Alshamrani et al. [6] used the LCA–LEED-NC v3 model to select the structure and types of building envelopes for new school buildings in Canada. They used the EAc1 credit (optimize energy performance, with a maximum of 19 points) from the EA category, all credits from the MR category (with a maximum of 13 points), and five environmental impacts—global warming potential, overall energy consumption, and air, water, and land emissions—from the LCA results (with a maximum score of five if each environmental impact is significantly reduced). Therefore, the LCA–LEED-NC v3 model has a maximum

score of 37. Alshamrani et al. [6] estimated seven structure–envelope alternatives: concrete, masonry, steel, wood, wood + masonry, steel + wood, and steel + masonry. As a result, a concrete building with minimum insulation can obtain the highest total LCA–LEED score (19), followed by masonry (17), while steel and steel–masonry receive the lowest score (14) [6]. The authors concluded that the LEED system needs careful restructuring to include LCA as an independent category with appropriate scores.

In 2022, it was found that in Californian cities, the certification strategy for LEED-CI v4 gold-certified office-space projects involved a trade-off between the LTc3 credit (access to quality transit) and the EAc6 credit (optimize energy performance) [7]; therefore, the LCA method must be used to select the best LEED certification strategy. In 2023, it was observed that in Manhattan, New York, the certification strategy for LEED-CI v4 gold-certified office-space projects included a trade-off between the EAc6 credit and three material-related credits: MRc2 (interiors life-cycle impact reduction), MRc5 (building product disclosure and optimization, material ingredients), and EQc2 (low-emitting materials) [8]. These two case studies show different LEED certification strategies. These analyses were possible due to a combination of several factors. On the one hand, the LEED-certified projects were in the same place, with one rating system, one rating version, and one level of certification; on the other hand, a suitable number of projects (sample size) was used to perform a non-parametric statistical analysis. The design of these two studies may be of interest to LEED stakeholders.

Recently, the author of this study compared the difference between Shanghai and California in terms of LEED-CI v4 gold-certified office-space projects [9]. The comparison revealed that the MR and EQ categories showed high levels of variability (interquartile range/median ratio = 0.57 for both categories) in LEED-CI v4 projects in Shanghai. This may be indirect evidence that there are two different strategies for LEED-CI certified projects [7]. However, the certification strategies for LEED-certified projects in Shanghai have not yet been explored. Thus, this study aims to (1) statistically define different LEED certification strategies, and (2) evaluate the environmental aspects of LEED strategies using LCA. The author focused on LEED-CI v4 gold-certified office spaces in Shanghai to minimize the impact of uncontrolled influences.

## 2. Materials and Methods

The study involved a two-step procedure: (1) an analysis of LEED-CI v4 gold-certified office-space certification strategies, and (2) LCA of the evaluated strategies. The first step involved (i) collecting LEED-CI v4 gold-certified office-space projects; (ii) sorting the collected projects into two groups, one with the lowest EAc6 credit achievement ($EA_{Low}$) and one with the highest EAc6 credit achievement ($EA_{High}$); and (iii) comparing points awarded for $EA_{Low}$ and $EA_{High}$ on the category and credit level. The second step involved (i) converting the credits of $EA_{Low}$ and $EA_{High}$ groups with different achievements into life cycle inventory (LCI) input for the production (P) and operational energy (OE) stages using the ecoinvent database [10], and (ii) converting the LCI output of the P and OE stages into a life cycle impact assessment (LCIA) using the ReCiPe2016 method [11] with two fuel source scenarios for the OE stage: 71.07% coal + 28.08% natural gas + 0.81% wind power (WP) + 0.04% photovoltaic (PV) [12] and 50% WP + 50% PV. Sections 2.1 and 2.2 provide detailed explanations of the first and second steps, respectively.

### 2.1. LEED-CI v4 Certification Strategies

2.1.1. Design of the Study

The design of the current study was based on the following assumptions: LEED data should be collected from one region, using the same rating system, the same version, the same certification level, and the same type of space, and with a suitable sample size. Such assumptions were applied to maximally reduce nondemonic intrusion, which is defined as "the impingement of chance events on an experiment in progress" [13].

The US Green Building Council (USGBC) and Green Building Information Gateway (GBIG) databases were used to collect 45 LEED-CI v4 gold-certified office-space projects in Shanghai [14,15]. These 45 projects were ranked from lowest to highest EAc6 credit achievement. The 15 lowest projects were identified as the $EA_{Low}$ group, and the 15 highest projects were identified as the $EA_{High}$ group.

### 2.1.2. Data Analysis
### Statistical Analysis

The Shapiro–Wilk test was used to check the assumption of normality using the *p*-value in terms of three-valued logic: seems to be positive (i.e., the normality assumption is met), seems to be negative (i.e., the normality assumption is not met), or judgment is suspended. More detailed information on the interpretation of *p*-values can be found in [16].

Parametric statistics include the mean ± standard deviation (SD), the SD/mean ratio, Cohen's *d* effect size with bias correction [17], and an unpaired *t*-test, while nonparametric statistics include the median, 25th–75th percentiles, interquartile range (IQR)/median ratio (IQR/M), Cliff's *δ* effect size [18], and exact Wilcoxon–Mann–Whitney (WMW) test.

If the LEED v4 data were associated with a binary scale, then the natural logarithm of the odds ratio ($ln\theta$) effect size was used [19], and Fisher's exact $2 \times 2$ test with Lancaster's mid-*p*-value was also used [20].

### Effect Size Procedure

In the present study, the effect size procedure was used to estimate substantive differences between the $EA_{Low}$ and $EA_{High}$ groups. The three tests used to estimate effect size are described below.

For the $ln\theta$ test, the Fleiss procedure (adding 0.5 to each observed frequency) was used if one of the proportions in the fourfold table was zero [21], as recommended by Haddock et al. [22]. However, this procedure may not work when the values in the other cells of the fourfold table are zero [22]. If $ln\theta$ is 0, there is no association between the two groups. The left and right limits of $ln\theta$ are infinity. The effect size is negligible if $|ln\theta| < 0.51$, small if $0.51 \leq |ln\theta| < 1.24$, medium if $1.24 \leq |ln\theta| < 1.90$, and large if $|ln\theta| \geq 1.90$ [23].

Cohen's *d* ranges from 0 to infinity. The effect size of Cohen's *d* is considered to be negligible if $|d| < 0.20$, small if $0.20 \leq |d| < 0.50$, medium if $0.50 \leq d < 0.80$, and large if $|d| \geq 0.80$ [24].

Cliff's *δ* ranges between −1 and +1. The effect size is negligible if $|\delta| < 0.147$, small if $0.147 \leq |\delta| < 0.33$, medium if $0.33 \leq |\delta| < 0.474$, and large if $|\delta| \geq 0.474$ [25].

### 2.1.3. *p*-Value Interpretation

Exact *p*-values were interpreted using three-valued logic [7].

### 2.2. LCA of LEED-CI v4 Certification

First, after evaluating the $EA_{Low}$ and $EA_{High}$ certification strategies, for the two groups of projects, the revealed high achievement in EAc6 and low achievement in EQc2 (low-emitting materials) and EQc8 (quality views) ($EAc6_{High}$–$EQc2\_EQc8_{Low}$) and low achievement in EAc6 and high achievement in EQc2 and EQc8 ($EAc6_{Low}$–$EQc2\_EQc8_{High}$) were converted from points to the corresponding LCI input; the ecoinvent database was used as a data source [10]. Then, LCI was converted to LCIA using the ReCiPe2016 method [11].

### 2.2.1. LCI: Functional Unit and System Boundary

The full LCA includes three main stages: production, operation, and demolition [26]. Thus, LCA was used to estimate the difference between two LEED certification strategies: $EAc6_{Low}$–$EQc2\_EQc8_{High}$ versus $EAc6_{High}$–$EQc2\_EQc8_{Low}$. In this context, the author used the P and OE stages. The requirements for EAc6 credit can be converted to measurable values at the OE stage to maintain cooling, heating, and lighting in the life cycle of the

building. The requirements for EQc2 and EQc8 credits can be converted to measurable values at the P stage to support production processes in the life cycle of a building.

Therefore, the FU was designated as follows: the production of building materials needed to ensure the requirements for EQc2 and EQc8 for 1 m$^2$ of the building area (the P stage) and the energy needed to provide heating, cooling, and lighting for 1 m$^2$ of the building area during 50 years of the building's lifetime (the OE stage).

EAc6 requires saving the building's OE stage needs for heating, cooling, and lighting; EQc2 involves using interior paints and coatings with low volatile organic compound (VOC) emissions; and EQc8 requires providing connections to the outdoors by providing vision glazing for 75% of the all regularly occupied floor area. Thus, the system boundary of LCA of the EAc6$_{High}$–EQc2_EQc8$_{Low}$ and EAc6$_{Low}$–EQc2_EQc8$_{High}$ certification strategies was modeled as an environmental benefit from the saved OE (EAc6; the OE stage) and environmental damage from paint for building interiors (EQc2) and glass, polyvinyl chloride (PVC), and concrete for exterior walls (EQc8; the P stage). Note that the demolition stage was excluded from the system boundary in this study because the environmental damage is significantly smaller in this stage than in the production and operational stages [27].

For both certification strategies, the ecoinvent database [10] was used to convert LCI input (kilograms of evaluated paint, glass, PVC, and concrete quantities and kilowatt-hours of OE) into LCI output (emission gases and waste). Table 2 shows the ecoinvent inventory data for the four most influential environmental impacts and the data sources for the P and OE stages of the EAc6$_{Low}$–EQc2_EQc8$_{High}$ and EAc6$_{High}$–EQc2_EQc8$_{Low}$ certification strategies applied in this study. The rest of the LCI results with all measures are presented in Appendix A, Table A1.

**Table 2.** LCI (most influential impacts and used data source) of EAc6$_{Low}$–EQc2_EQc8$_{High}$ and EAc6$_{High}$–EQc2_EQc8$_{Low}$ certification strategies.

| LCA Stage (Credit) | Material/Energy Input | GW (kg CO$_2$ eq) | HCT (kg 1,4-DCB) | Hn-CT(kg 1,4-DCB) | TE (kg 1,4-DCB) | Ecoinvent v3.2 Data Source [10] |
|---|---|---|---|---|---|---|
| OE (EAc6) | OE: 71.07% coal + 28.08% natural gas + 0.81% WP + 0.04% PV (1 kWh/m$^2$·50 years) | 1.230 | 0.0191 | 0.0222 | 0.4163 | Electricity, hard coal, at plant/CN Electricity, at refinery/CH Electricity, wind power/CN Electricity photovoltaic/CN |
| | OE: 50% WP and 50% PV (1 kWh/m$^2$·50 years) | 0.0011 | 0.0001 | 0.0004 | 0.004 | |
| P (EQc8) | Wall: glass (1 kg) | 0.393 | 0.013 | 0.024 | 1.360 | Flat glass, uncoated, at plant/RER |
| | Wall: PVC (1 kg) | 2.05 | 0.093 | 0.025 | 0.998 | Polyvinyl chloride, at plant/RER |
| | Wall: concrete (1 kg) | 0.938 | 0.001 | 0.002 | 0.081 | Concrete, exacting, at plant/CH |
| P (EQc2) | Paint: eco-friendly (1 kg) | 5.010 | 0.115 | 3.570 | 4.620 | Alkyd paint, without water/RER |
| | Paint: typical (1 kg) | 5.700 | 0.178 | 4.480 | 13.00 | Alkyd paint, without solvent/RER |

OE, operational energy; P, production; WP, wind power; PV, photovoltaic; PVC, polyvinyl chloride; GW, global warming; HCT, human carcinogenic toxicity; Hn-CT, human noncarcinogenic toxicity; TE, terrestrial ecotoxicity; CN, China; CH, Switzerland; RER, France. EAc6: optimize energy performance; EQc8: L quality views; EQc2: low-emitting materials.

### 2.2.2. LCIA: ReCiPe Method

ReCiPe2016 LCIA evaluates 22 environmental impacts, including global warming potential, fossil resource scarcity, human toxicity, terrestrial ecotoxicity, and water consumption, and converts these into damage to human health, ecosystem quality, and resources [11].

This method allows us to consider human health, ecosystem quality, and resource-based environmental damage from living pollutants on three time horizons: individualist (I; 20 years), hierarchist (H; 100 years), and egalitarian (E; infinite) [11]. In this study, the same average (A) weightings for human health, ecosystem quality, and resources (40%, 40%, and 20%, respectively) were applied to the I, H, and E time horizons. As such, the I/A, H/A, and E/A methodological options were used for LCIA of the identified LEED-CI v4 certification strategies.

## 3. Results and Discussion

### 3.1. Choice between Parametric and Nonparametric Statistics

Table 3 shows that the assumption of normality was met in both the $EA_{Low}$ and $EA_{High}$ groups ($p$ = 0.2821 and 0.2876, respectively). Further, the parametric and nonparametric descriptive statistics and inferential statistics showed similar results. Although the normality assumption was met in both groups, indicating a preference for parametric methods, nonparametric procedures were preferred because LEED v4 data do not refer to interval variables [28,29].

**Table 3.** LEED total points, Shanghai.

| Group | Shapiro–Wilk Test (*p*-Value) | Mean ± SD (SD/Mean Ratio) | Median, 25th–75th Percentiles (IQR/Median Ratio) | Parametric Cohen's *d* and *t*-Test (*p*-Value) | Nonparametric Cliff's $\delta$ and WMW Test (*p*-Value) |
|---|---|---|---|---|---|
| $EA_{Low}$ | 0.2821 | 63.07 ± 2.28 (0.04) | 63.0 61.0–65.0 (0.06) | −0.96 (0.0109) [a] | −0.54 (0.0097) [a] |
| $EA_{High}$ | 0.2876 | 66.00 ± 3.48 (0.05) | 65.0 63.5–67.8 (0.07) | | |

Note: [a] Difference between two groups seems to be negative.

Table 4 shows that for the IP, LT, WE, MR, IO, and RP categories in the $EA_{Low}$ and $EA_{High}$ groups, and for the EA and EQ categories in one of the groups, the normality assumption is not met. Consequently, in this context, nonparametric statistics should be used to compare the two groups.

**Table 4.** Checking assumption of normality at category level in Shanghai.

| Category | Maximum Points | Group | *p*-Value |
|---|---|---|---|
| Integrative process (IP) | 2 | $EA_{Low}$ | <0.0001 |
| | | $EA_{High}$ | <0.0001 |
| Location and transportation (LT) | 18 | $EA_{Low}$ | 0.0005 |
| | | $EA_{High}$ | 0.0010 |
| Water efficiency (WE) | 12 | $EA_{Low}$ | 0.0009 |
| | | $EA_{High}$ | 0.0003 |
| Energy and atmosphere (EA) | 38 | $EA_{Low}$ | 0.0139 |
| | | $EA_{High}$ | 0.2284 |
| Materials and resources (MR) | 13 | $EA_{Low}$ | 0.0423 |
| | | $EA_{High}$ | 0.0256 |
| Indoor environmental quality (EQ) | 17 | $EA_{Low}$ | 0.0319 |
| | | $EA_{High}$ | 0.3029 |
| Innovation (IO) | 6 | $EA_{Low}$ | 0.0025 |
| | | $EA_{High}$ | 0.0011 |
| Regional priority (RP) | 4 | $EA_{Low}$ | 0.0011 |
| | | $EA_{High}$ | 0.0009 |

Table 5 shows that for the selected LEED credits in either of the two groups or in both groups, the normality assumption does not hold. Therefore, in this context, nonparametric statistics should be used.

**Table 5.** Checking assumption of normality at credit level in Shanghai.

| Category | Maximum Points | Group | *p*-Value |
|---|---|---|---|
| Enhanced refrigerant management (EAc4) | 1 | $EA_{Low}$ | <0.0001 |
| | | $EA_{High}$ | 0.0001 |
| Optimize energy performance (EAc6) | 25 | $EA_{Low}$ | 0.0001 |
| | | $EA_{High}$ | 0.0622 |
| Low-emitting materials (EQc2) | 3 | $EA_{Low}$ | 0.0124 |
| | | $EA_{High}$ | <0.0001 |
| Quality views (EQc8) | 1 | $EA_{Low}$ | <0.0001 |
| | | $EA_{High}$ | <0.0001 |

The author concluded that (1) the LEED data are associated with a binary scale, an ordinal scale, or interval variables with relatively few values, and (2) the normality assumption is generally not met for the LEED data. In this context, nonparametric statistics should be used instead of parametric statistics.

*3.2. Two Strategies for Obtaining LEED Gold Certification*

Table 6 presents a statistical analysis of LEED-CI v4 gold-certified office-space projects in Shanghai, China, with $EA_{Low}$ and $EA_{High}$ achievement at the category level. It can be seen that among the five main categories, LT, WE, and EA showed low variability and high achievement, while MR and EQ showed high variability and low achievement. These results were found for both $EA_{Low}$ and $EA_{High}$ groups. Similar results were previously reported from an evaluation of LEED-CI v4 gold-certified office-space projects in Shanghai without taking into account the possibility of diversifying the project certification strategies [9].

**Table 6.** $EA_{Low}$ versus $EA_{High}$ achievement, Shanghai.

| Category | Maximum Points | Group | Median, 25th–75th Percentiles | IQR/M | Cliff's $\delta$ | *p* |
|---|---|---|---|---|---|---|
| Integrative process (IP) | 2 | $EA_{Low}$ | 2.0, 2.0–2.0 | 0.00 | 0.18 | 0.2648 |
| | | $EA_{High}$ | 2.0, 0.0–2.0 | 1.00 | | |
| Location and transportation (LT) | 18 | $EA_{Low}$ | 17.0, 17.0–18.0 | 0.06 | 0.22 | 0.3251 |
| | | $EA_{High}$ | 17.0, 17.0–17.8 | 0.04 | | |
| Water efficiency (WE) | 12 | $EA_{Low}$ | 12.0, 8.5–12.0 | 0.29 | 0.08 | 0.6939 |
| | | $EA_{High}$ | 10.0, 10.0–12.0 | 0.20 | | |
| Energy and atmosphere (EA) | 38 | $EA_{Low}$ | 13.0, 12.3–15.0 | 0.21 | −0.98 | <0.0001 |
| | | $EA_{High}$ | 20.0, 18.0–24.0 | 0.30 | | |
| Materials and resources (MR) | 13 | $EA_{Low}$ | 4.0, 3.0–5.0 | 0.50 | 0.28 | 0.1736 |
| | | $EA_{High}$ | 2.0, 2.0–5.0 | 1.50 | | |
| Indoor environmental quality (EQ) | 17 | $EA_{Low}$ | 8.0, 7.0–11.0 | 0.50 | 0.52 | 0.0133 |
| | | $EA_{High}$ | 6.0, 4.0–8.8 | 0.79 | | |
| Innovation (IO) | 6 | $EA_{Low}$ | 5.0, 5.0–6.0 | 0.20 | 0.42 | 0.0382 |
| | | $EA_{High}$ | 5.0, 4.3–5.0 | 0.15 | | |
| Regional priority (RP) | 4 | $EA_{Low}$ | 3.0, 3.0–3.0 | 0.00 | −0.25 | 0.2330 |
| | | $EA_{High}$ | 3.0, 3.0–4.0 | 0.33 | | |
| LEED total | 110 | $EA_{Low}$ | 63.0 61.0–65.0 | 0.06 | −0.54 | 0.0097 |
| | | $EA_{High}$ | 65.0 63.5–67.8 | 0.07 | | |

IQR/M, interquartile range/median ratio.

Table 6 shows that the $EA_{Low}$ group scored significantly lower than the $EA_{High}$ group in the EA category ($\delta = -0.98$, $p < 0.0001$). In contrast, the $EA_{Low}$ group scored significantly higher than the $EA_{High}$ group in both the EQ and IO categories ($\delta = 0.52$, $p = 0.0133$ and $\delta = 0.42$, $p = 0.0382$, respectively). Thus, these two groups can be renamed $EA_{Low}$–$EQ_{High}$ and $EA_{High}$–$EQ_{Low}$. The IO category was omitted because it is not possible to convert its achievement to LCA.

In addition, it can be seen that the $EA_{Low}$–$EQ_{High}$ group did not perform as well as the $EA_{High}$–$EQ_{Low}$ group in terms of total LEED scores ($\delta = -0.54$, $p = 0.0097$). This means that the use of the $EA_{High}$–$EQ_{Low}$ strategy is associated with more sustainable building design.

Table 7 presents a statistical analysis of the LEED-CI data in Shanghai, China, with $EA_{Low}$–$EQ_{High}$ and $EA_{High}$–$EQ_{Low}$ achievements at the credit level. The credits show the category in which the difference between the two groups seems to be positive. Notably, the $EA_{Low}$–$EQ_{High}$ group scored significantly lower than the $EA_{High}$–$EQ_{Low}$ group in the EAc6 credit ($\delta = -1.00$, $p = <0.0001$). In contrast, the $EA_{Low}$–$EQ_{High}$ group scored significantly higher than the $EA_{High}$–$EQ_{Low}$ group in the EAc4 (enhanced refrigerant management), EQc2, and EQc8 credits ($ln\theta = 2.51$, $p = 0.0269$; $\delta = 0.39$, $p = 0.0446$; and $ln\theta = 1.70$, $p = 0.0233$, respectively).

**Table 7.** Difference between $EA_{Low}$ and $EA_{High}$ seems to be positive in Shanghai.

| Credit | Maximum Points | Group | Median, 25th–75th Percentiles | IQR/M | Cliff's $\delta$/$ln\theta$ | $p$ |
|---|---|---|---|---|---|---|
| Enhanced refrigerant management [2] (EAc4) | 1 | $EA_{Low}$–$EQ_{High}$ | 1.0, 1.0–1.0 | 0.00 | 2.51 | 0.0269 |
| | | $EA_{High}$–$EQ_{Low}$ | 1.0, 0.0–1.0 | 1.00 | | |
| Optimize energy performance (EAc6) [1] | 25 | $EA_{Low}$–$EQ_{High}$ | 8.0, 6.3–8.0 | 0.22 | −1.00 | <0.0001 |
| | | $EA_{High}$–$EQ_{Low}$ | 14.0, 11.3–20.3 | 0.64 | | |
| Low-emitting materials (EQc2) [1] | 3 | $EA_{Low}$–$EQ_{High}$ | 1.0, 0.0–2.0 | 2.00 | 0.39 | 0.0446 |
| | | $EA_{High}$–$EQ_{Low}$ | 0.0, 0.0–0.0 | NaN | | |
| Quality views (EQc8) [2] | 1 | $EA_{Low}$–$EQ_{High}$ | 1.0, 0.3–1.0 | 0.75 | 1.70 | 0.0233 |
| | | $EA_{High}$–$EQ_{Low}$ | 0.0, 0.0–1.0 | Inf | | |

Notes: [1] Cliff's $\delta$ and exact Wilcoxon–Mann–Whitney test with two-tailed $p$-value were used to estimate difference between $EA_{Low}$–$EQ_{High}$ and $EA_{High}$–$EQ_{Low}$. [2] Natural logarithm of odds ratio ($ln\theta$) and Fisher's exact test $2 \times 2$ table with two-tailed mid-$p$-value were used to estimate difference between $EA_{Low}$–$EQ_{High}$ and $EA_{High}$–$EQ_{Low}$. IQR/M, interquartile range/median ratio; NaN, not a number; Inf, result of numerical calculation that is mathematically infinite.

Tables 8 and 9 show the EAc6, EAc4, EQc2, and EQc8 scores for the two LEED certification strategies, $EA_{Low}$-$EQ_{High}$ and $EA_{High}$-$EQ_{Low}$, respectively. The resulting median credit scores were used to perform LCA of the two strategies.

**Table 8.** $EA_{Low}$–$EQ_{High}$ strategies: points awarded for credits.

| | Project | | EAc6 | EAc4 | EQc2 | EQc8 |
|---|---|---|---|---|---|---|
| No | Name | Address | | Achieved | Points | |
| 1 | HKS Shanghai Office | Changle Road, Xuhui District | 0 | 1 | 0 | 1 |
| 2 | Steelcase Worklife Shanghai | 39/F, HKRI Tower 1 | 5 | 1 | 2 | 1 |
| 3 | Bank of East Asia Tower | 299 Si Chuan Road Central, Huang Pu | 6 | 1 | 2 | 0 |
| 4 | Hang Seng Bank Headquarters | No. 1000 Lujiazui Ring Road | 6 | 1 | 3 | 1 |
| 5 | Allergan Shanghai Office | 58th Floor, Plaza 66, 1266 West Nanjing | 7 | 1 | 0 | 0 |
| 6 | China Life Office | 88 Yincheng Rd. | 7 | 1 | 1 | 1 |
| 7 | Haworth Kerry Center Showroom | 32/F Tower 1, JingAn Kerry Center | 8 | 1 | 0 | 1 |

**Table 8.** *Cont.*

| Project | | | EAc6 | EAc4 | EQc2 | EQc8 |
|---|---|---|---|---|---|---|
| No | Name | Address | Achieved Points | | | |
| 8 | SIP Main Lobby | Hongkou District | 8 | 1 | 1 | 0 |
| 9 | Alliance Bernstein Shanghai Office | 16/F HKRI Centre Two | 8 | 1 | 0 | 1 |
| 10 | Two Sigma Shanghai Tower | No. 501 Middle Yin Cheng Road, Pudong | 8 | 0 | 1 | 1 |
| 11 | Sirio Shanghai Office | 1139 Changning Road | 8 | 1 | 2 | 1 |
| 12 | KKR Shanghai Office | 43/F, HKRI Centre One, HKRI Taikoo Hui | 8 | 1 | 2 | 1 |
| 13 | Hilton Bund Center 46F Office | 222 Yan'an East Road, Huangpu District | 8 | 1 | 3 | 1 |
| 14 | ZhangJiang CITI Bank Office | Zhangjiang | 8 | 1 | 0 | 0 |
| 15 | LinkedIn Shanghai Office | 999 Huaihai Middle Road, Huangpu | 8 | 1 | 0 | 1 |
| | *Median* | | *8* | *1* | *1* | *1* |

Notes: EAc6: optimize energy performance; EAc4: enhanced refrigerant management; EQc2: low-emitting materials; EQc8: quality views. Bold italic font indicates median of credit scores used to conduct LCA to evaluate LEED certification strategies.

**Table 9.** $EA_{High}$–$EQ_{Low}$ strategies: points awarded for credits.

| Project | | | EAc6 | EAc4 | EQc2 | EQc8 |
|---|---|---|---|---|---|---|
| No | Name | Address | Achieved Points | | | |
| 1 | Zofund Project | Shanghai, 200120, CN | 10 | 1 | 0 | 0 |
| 2 | BV CPS Shanghai | 248 Guanghua Rd, Minhang Qu | 10 | 0 | 0 | 0 |
| 3 | Khazanah National Shang Office | 49/F, 2IFC, No. 8 Century Avenue | 11 | 1 | 2 | 1 |
| 4 | China Life 57th floor | No. 88, Yincheng Road, Pudong District | 11 | 0 | 0 | 1 |
| 5 | Adidas Shanghai Headquarters Office | No. 160, Gongcheng Road | 12 | 1 | 1 | 0 |
| 6 | Bulgari Shanghai Office Project | 1266 Nanjing W Rd, Nan Jing Xi Lu | 12 | 0 | 0 | 0 |
| 7 | Kering Offices Garden Square | 29 F, No. 968 Beijing West Road | 13 | 0 | 0 | 0 |
| 8 | Unity Shanghai Office | Sinar Mas Plaza, No. 501 Dongdaming Rd | 14 | 1 | 0 | 0 |
| 9 | Apple SC2 T4 L9 | Century Metropolis (Tower 4) | 15 | 1 | 0 | 0 |
| 10 | Apple SC2 T4 L10 | No. 288 Fushan Road | 17 | 0 | 0 | 0 |
| 11 | Gensler Shanghai Office | One Museum Place, 3/F | 18 | 0 | 3 | 1 |
| 12 | L'Oreal TR China Hub Office | No. 8 Shi Ji Da Dao | 21 | 1 | 0 | 0 |
| 13 | Shanghai Swiss Re Consultancy | 179 Weifang Rd, Pudong Xinqu | 22 | 1 | 0 | 1 |
| 14 | POLESTAR Shanghai Office | No. 555, Dong Da Ming Road | 23 | 1 | 0 | 1 |
| 15 | JPMC Shanghai Tower Project Phase2 | F45-48, No.501 Yincheng Middle Road | 24 | *0* | *0* | *0* |
| | *Median* | | *14* | *1* | *0* | *0* |

Notes: EAc6: optimize energy performance; EAc4: enhanced refrigerant management; EQc2: low-emitting materials; EQc8: quality views. Bold italic font indicates median of credit scores used to conduct LCA to evaluate LEED certification strategies.

### 3.3. LCA of Identified LEED Gold-Certified Strategies

3.3.1. Preliminary Results: From Credits to Environmental Benefit/Damage: Life-Cycle Inventory

According to the median results of achieved points, the difference between the two strategies was in the achievement of EAc6, EQc2, and EQc8, with respectively 8 points, 1 point, and 1 point for $EA_{Low}$–$EQ_{High}$ (Table 8) and 14 points, 0 points, and 0 points for $EA_{High}$–$EQ_{Low}$ (Table 9), whereas the median result for EAc4 was the same for both

strategies, 1 point (Tables 8 and 9). According to the LCA methodology, only differences between compared alternatives need to be evaluated [26]. Therefore, the LCA results of $EAc6_{Low}$–$EQc2\_EQc8_{High}$ and $EAc6_{High}$–$EQc2\_EQc8_{Low}$ were evaluated.

*Converting the requirements of EAc6 to the OE stage.* According to Han et al. [30], a typical office building in Shanghai consumes 126 kWh/m$^2$ of OE (base OE intensity). The $EAc6_{Low}$–$EQC2\_EQc8_{High}$ certification strategy received 8 points in EAc6 (Table 8), which corresponds to 6% saved OE [5], whereas the $EAc6_{High}$–$EQc2\_EQc8_{Low}$ certification strategy received 14 points (Table 9), which corresponds to 11% saved OE [5]. Thus, the environmental benefit of OE savings was evaluated as 6% and 11% of the base OE intensity per 50 years of the building's lifetime for $EAc6_{Low}$–$EQc2\_EQc8_{High}$ and $EAc6_{High}$–$EQc2\_EQc8_{Low}$ certification strategies, respectively (Table 10).

**Table 10.** Input used for LCA of $EAc6_{Low}$–$EQc2\_EQc8_{High}$ and $EAc6_{High}$–$EQc2\_EQc8_{Low}$ certification strategies.

| LCA Stage (Credit) | Input | $EAc6_{Low}$–$EAc4\_EQc2\_EQc8_{High}$ | $EAc6_{High}$–$EAc4\_EQc2\_EQc8_{Low}$ |
|---|---|---|---|
| OE (EAc6) | OE: 71.07% coal + 28.08% natural gas + 0.81% WP + 0.04% PV (kWh/m$^2$·50 years) <br> OE: 50% WP and 50% PV (kWh/m$^2$·50 years) | 126 kWh/m$^2$·0.06·50 years = $-378$ | 126 kWh/m$^2$·0.11·50 years = $-693$ |
| P (EQc8) | Wall: glass (kg) | $(0.45$ m$^2$·0.009 m·2500 kg/m$^2)$·2 = 20.3 | $(0.15$ m$^2$·0.009 m·2500 kg/m$^2)$·2 = 6.8 |
| | Wall: PVC (kg) | $(0.45$ m$^2$·0.1·0.04 m·1030 kg/m$^2)$·2 = 3.7 | $(0.15$ m$^2$·0.1·0.04 m·1030 kg/m$^2)$·2 = 1.2 |
| | Wall: concrete (kg) | 0.15 m$^2$·0.2 m·2400 kg/m$^2$ = 72 | 0.45 m$^2$·0.2 m·2400 kg/m$^2$ = 216 |
| P (EQc2) | Paint: eco-friendly (kg) <br> Paint: typical (kg) | 0.15 m$^2$·0.35 kg·4 = 0.2 <br> - | - <br> 0.45 m$^2$·0.35 kg·4 = 0.6 |

OE, operational energy; P, production; WP, wind power; PV, photovoltaic; PVC, polyvinyl chloride. EAc6: optimize energy performance; EQc8: quality views; EQc2: low-emitting materials.

*Converting the requirements of EQc8 to P stage.* EQc8 requires that 75% of all regularly occupied floor areas have quality views [5]. In this credit, the $EAc6_{Low}$–$EQc2\_EQc8_{High}$ and $EAc6_{High}$–$EQc2\_EQc8_{Low}$ certification strategies received 1 and 0 points, respectively (Tables 8 and 9). This study is limited to a typical closed floor plan office building [31]. This type of office building usually includes meeting rooms, break rooms, and other non-permanent utility spaces in the interior area; so they have interior walls without outside windows. Separate offices are located along the external walls; so everyone has a window. According to this type of office building, 100% of all regularly occupied floor areas can have quality views. Thus, achievement points of EQc8 depend on window size for the separate offices: large window allows quality views; small windows cannot allow quality views. Note that meeting rooms, break rooms, and other non-permanent utility spaces in the interior area are the same for $EAc6_{Low}$–$EQc2\_EQc8_{High}$ and $EAc6_{High}$–$EQc2\_EQc8_{Low}$ certification strategies, whereas window sizes are different for these strategies: large windows for $EAc6_{Low}$–$EQc2\_EQc8_{High}$ and small windows for $EAc6_{High}$–$EQc2\_EQc8_{Low}$. Only different components/materials should be accounted for LCA [26]. This means that only the material quantities of external wall, including windows, can serve as a metric to convert the requirements of EQc8 to the LCA.

According to the literature, in a typical closed floor plan office building, the area of the external wall comprises 60% of the floor area [31]. Thus, 0.6 m$^2$ of wall per FU (1 m$^2$ of floor) was evaluated as 0.45 m$^2$ of window + 0.15 m$^2$ of concrete wall for $EAc6_{Low}$–$EQc2\_EQc8_{High}$ and 0.15 m$^2$ of window + 0.45 m$^2$ of concrete wall for $EAc6_{High}$–$EQc2\_EQc8_{Low}$. The windows comprised double clear glass panes 3 and 6 mm thick and a polyvinyl chloride (PVC) frame 4 cm thick around 10% of the window area, and they were replaced two times during the building's lifetime (50 years). Using these figures, the resulting quantities of wall-related concrete, glass, and PVC were evaluated (Table 10).

*Converting the requirements of EQc2 to P stage.* EQc2 requires the use of paint with low VOCs [5]. In this credit, EAc6$_{Low}$–EQc2_EQc8$_{High}$ and EAc6$_{High}$–EQc2_EQc8$_{Low}$ strategies received 1 and 0 points, respectively (Tables 6 and 7). Covering 1 m$^2$ of wall requires 0.35 kg of paint [32]. Thus, 0.15 m$^2$ of concrete wall (EAc6$_{Low}$–EQc2_EQc8$_{High}$, EQc8) was covered with eco-friendly paint and 0.45 m$^2$ of concrete wall (EAc6$_{High}$–EQc2_EQc8$_{Low}$, EQc8) was covered with typical paint. The paint was replaced four times during the building's lifetime (50 years). Using these figures, the resulting quantities of eco-friendly and typical paint were evaluated (Table 10).

Table 10 shows the input of OE kilowatts (OE stage, EAc6) and kilograms of materials (P stage, EQc2 and EQc8) that present environmental benefits (negative values) and damage (positive values), respectively. To calculate the LCI of the evaluated certification strategies, for the input energy and material records (Table 10), the ecoinvent data presented in Table 2 were modeled on the SimaPro platform [10].

### 3.3.2. From Credits to Environmental Benefit/Damage: A Life-Cycle Impact Assessment

Figure 1 shows the environmental benefit and damage associated with two gold-certified strategies, EAc6$_{Low}$–EQc2_EQc8$_{High}$ and EAc6$_{High}$–EQc2_EQc8$_{Low}$. As expected, in the P stage of materials used to achieve LEED points in EQc2 and EQc8, the EAc6$_{Low}$–EQc2_EQc8$_{High}$ strategy incurred less environmental damage than EAc6$_{High}$–EQc2_EQc8$_{Low}$. This result was obtained via three perspectives on the importance of the environmental problem: I/A, H/A, and E/A.

In the OE stage of saving energy to achieve LEED points in EAc6, EAc6$_{High}$–EQc2_EQc8$_{Low}$ resulted in greater environmental benefits than EAc6$_{Low}$–EQc2_EQc8$_{High}$. This result was confirmed for the three perspectives, I/A, H/A, and E/A. However, the difference between EAc6$_{High}$–EQc2_EQc8$_{Low}$ and EAc6$_{Low}$–EQc2_EQc8$_{High}$ was more significant for the scenario with 71.07% coal + 28.08% natural gas + 0.81% WP + 0.04% PV fuel sources than for the scenario with 50% WP + 50% PV fuel sources. Moreover, greater environmental benefits resulted from the use of 71.07% coal + 28.08% natural gas + 0.81% WP + 0.04% PV (currently used fuel sources) than from 50% WP + 50% PV (hypothetical future fuel sources). The influence of the transition from fossil to renewable energy sources on decreasing the OE stage is well known and referenced throughout the literature [7,33].

Therefore, the total P + OE results were completely different in the two fuel source scenarios. In the case of 71.07% coal + 28.08% natural gas + 0.81% WP + 0.04% PV, the environmental benefit of both certification strategies was obvious for all three methodological options (I/A, H/A, and E/A), with a predominant effect of EAc6$_{High}$–EQc2_EQc8$_{Low}$ on EAc6$_{Low}$–EQc2_EQc8$_{High}$. However, in the case of 50% WP + 50% PV, the I/A, H/A, and E/A results confirmed environmental damage. This is due to the dominance of the P stage over the decreased OE stage when using renewable fuel sources. Moreover, the total environmental influence of P + OE on the EAc6$_{High}$–EQc2_EQc8$_{Low}$ and EAc6$_{Low}$–EQc2_EQc8$_{High}$ strategies was completely different for the present scenario of a mix of fossil and renewable fuel sources, with the former favored as more beneficial. Meanwhile, the total environmental influence of P + OE was very similar for both strategies in the hypothetical future scenario of renewable fuel sources.

Other authors have also highlighted the increased influence of the P stage when renewable fuel sources are used for OE. In this respect, Giordano et al. [34] indicated the importance of the P stage for near-zero-energy buildings (nZEBs) that use renewable fuel sources. Lessard et al. [35] evaluated the LCA of an office building located in Canada, where almost all OE is produced with renewable fuel sources, and they concluded that the P stage overcame the OE stage.

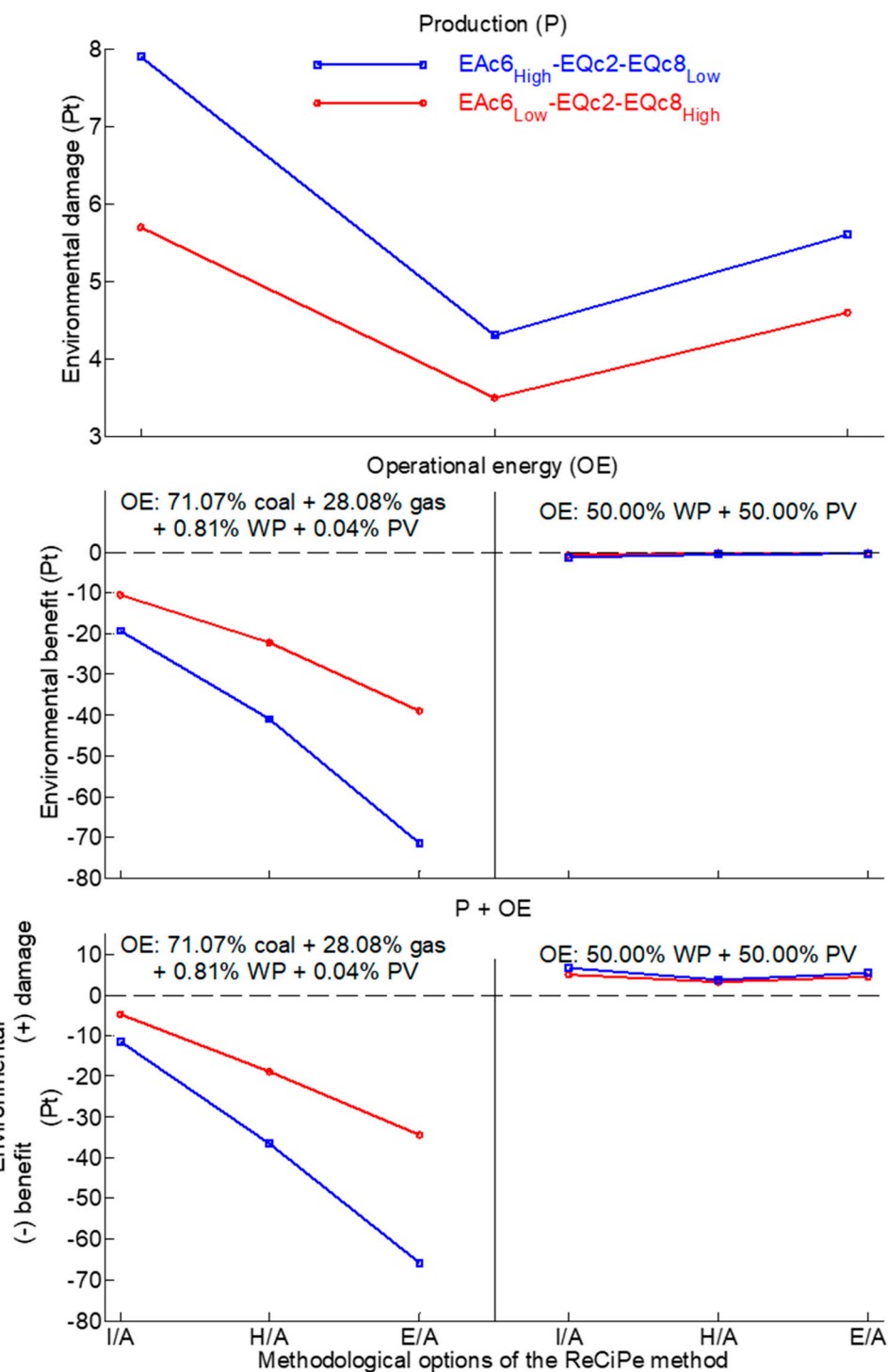

**Figure 1.** Environmental benefit/damage of EAc6$_{Low}$–EQc2_EQc8$_{High}$ and EAc6$_{High}$–EQc2_EQc8$_{Low}$ gold-certified strategies evaluated using ReCiPe2016 methodological options.

Thus, when the LEED certification strategy includes the EAc6 credit along with other material-related credits (EQc2 and EQc8), it is highly dependent on the fuel source for the OE stage.

## 4. Conclusions

This study focused on the certification strategies of LEED-CI v4 gold-certified office-space projects in Shanghai, China. First, the strategies were analyzed at the category

and credit level, and then, they were evaluated using LCA. The following conclusions were reached.

At the category level, LEED-CI v4 gold-certified office-space projects used two certification strategies: $EA_{Low}$–$EQ_{High}$ and $EA_{High}$–$EQ_{Low}$. Even though both strategies received the same gold certification, $EA_{High}$–$EQ_{Low}$ was found to be more sustainable than $EA_{Low}$–$EQ_{High}$. This is due to the overall LEED achievement, which was significantly higher for the former than the latter strategy. At the credit level, EAc6 (optimize energy performance), Eqc2 (low-emitting materials), and EQc8 (quality views) were identified as the credits responsible for the certification strategies $EAc6_{Low}$–$EQc2\_EQc8_{High}$ and $EAc6_{High}$–$EQc2\_EQc8_{Low}$.

The LCA results show that EAc6 credit achievement was associated with environmental benefits resulting from a building's operating energy, while EQc2 and EQc8 credits were associated with environmental damage from the use of building materials. Thus, according to the ReCiPe2016 results, at the P stage, the use of the $EAc6_{Low}$–$EQc2\_EQc8_{High}$ strategy caused less damage to the environment, and at the OE stage, the use of the $EAc6_{High}$–$EQc2\_EQc8_{Low}$ strategy brought about more environmental benefits. However, the OE (benefit) stage had a predominant effect compared to the P (damage) stage. Based on the LCA (P + OE), $EAc6_{High}$–$EQc2\_EQc8_{Low}$ has been recognized as a more sustainable certification strategy for LEED-CI v4 gold-certified office-space projects. However, this strategy was very sensitive to the source of fuel used to meet the needs of the OE stage. When a combination of fossil and renewable fuels was used, the $EAc6_{High}$–$EQc2\_EQc8_{Low}$ strategy delivered more environmental benefits than the $EAc6_{Low}$–$EQc2\_EQc8_{High}$ strategy; moreover, when a combination of renewable fuels was used, both strategies had nearly the same impact on the environmental performance of LEED projects.

It can be concluded that LEED-certified buildings that have the same level of certification, but use different certification strategies can bring about different degrees of environmental damage/benefit. The problem of choosing a LEED-certified strategy is exacerbated when EAc6 (optimize energy performance) credit achievement is considered without taking into account the fuel sources used during the building's operating energy stage. This means that when choosing the best LEED-certified strategy, it is essential to perform an LCA in order to achieve sustainable development.

## 5. Limitation

To better understand green building strategies using LCA, it is necessary to study LEED projects from other systems, such as LEED-EB (existing building), LEED-NC (new build and major renovation), and so on. The certification of LEED strategies may also depend on the level of LEED certification, the urban infrastructure, and the building technology. Particular attention should be paid to the relationship between green building strategies and climate change.

In this study, the author used the interior of a typical closed floor plan office building to assess the environmental impact/benefit of LEED-CI certified projects. This approach was chosen due to the lack of information about the internal plan for each LEED project. Future research should explore the impact of building design on the choice of green building certification strategy using life cycle assessment.

## 6. Recommendation

Today, evaluating the construction and operation of green buildings in terms of LCA is moving from science to practice. Therefore, LCA should be more deeply integrated into the LEED system, not just at the materials and resources level, as is the case in LEED version 4, but through changes in local green policy.

**Funding:** This research received no external funding.

**Informed Consent Statement:** Not applicable.

**Data Availability Statement:** Publicly available datasets were analyzed in this study. The data can be found here: https://www.usgbc.org/projects (USGBC Projects Site) (accessed on 20 August 2022) and http://www.gbig.org (GBIG Green Building Data) (accessed on 20 August 2022).

**Conflicts of Interest:** The author declares no conflict of interest.

## Appendix A

**Table A1.** LCI of EAc6$_{Low}$–EQc2_EQc8$_{High}$ and EAc6$_{High}$–EQc2_EQc8$_{Low}$ certification strategies.

| LCA Stage (Credit) | Material/Energy Input | SOD (kg CFC11 eq) | IR (kBq Co-60 eq) | OzF (kg NOx eq) | FPMF (kg PM2.5 eq) | TA (kg SO$_2$ eq) | LU (m$^2$a crop eq) | WC (m$^3$) |
|---|---|---|---|---|---|---|---|---|
| OE (EAc6) | OE: 71.07% coal + 28.08% natural gas + 0.81% WP + 0.04% PV (1 kWh/m$^2$·50 years) | $9.1 \times 10^{-8}$ | 0.002 | 0.003 | 0.003 | 0.008 | 0.015 | 0.066 |
| | OE: 50% WP and 50% PV (1 kWh/m$^2$·50 years) | $7.9 \times 10^{-11}$ | $2.0 \times 10^{-4}$ | $1.8 \times 10^{-6}$ | $1.1 \times 10^{-6}$ | $1.9 \times 10^{-6}$ | $1.3 \times 10^{-5}$ | $1.7 \times 10^{-2}$ |
| P (EQc8) | Wall: glass (1 kg) | $1.8 \times 10^{-7}$ | 0.073 | 0.004 | 0.002 | 0.007 | 0.013 | 1.320 |
| | Wall: PVC (1 kg) | $5.9 \times 10^{-7}$ | 0.006 | 0.005 | 0.001 | 0.004 | 0.001 | 0.562 |
| | Wall: concrete (1 kg) | $1.1 \times 10^{-8}$ | 0.138 | 0.002 | 0.006 | 0.001 | 0.001 | 0.271 |
| P (EQc2) | Paint: eco-friendly (1 kg) | $3.8 \times 10^{-6}$ | 0.024 | 0.009 | 0.013 | 0.042 | 0.648 | 0.107 |
| | Paint: typical (1 kg) | $5.5 \times 10^{-6}$ | 0.366 | 0.012 | 0.016 | 0.047 | 2.270 | 0.124 |

OE, operational energy; P, production; PV, photovoltaic; WP, wind power; PVC, polyvinyl chloride; SOD, stratospheric ozone depletion; IR, ionizing radiation; OzF, ozone formation; FPMF, fine particulate matter formation; TA, terrestrial acidification; LU, land use; WC, water consumption. EAc6: optimize energy performance; EQc8: quality views; EQc2, low-emitting materials.

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
