# Peer review of "Life-Cycle Assessment of LEED-CI v4 Projects in Shanghai, China: A Case Study"

_sustainability, doi:10.3390/su15075722_

Round 1

Reviewer 1 Report

In this manuscript the author appears to be attempting to contrast the merits of two different LEED certification strategies. While a critical evaluation of certification strategies is an important part of ensuring that they are meeting objectives of the program conferring the certification, the manuscript fails to establish validity of the methodological approach being used to perform the evaluation. Critically, I note that it appears that an entire section of the manuscript (Section 3.2.1) may have been inadvertently left out of the manuscript and this might be the cause of the lack of clarity. In addition to ensuring the manuscript is complete, the introduction could also use more detail to explain how the author is applying LCA in the LEED context since there doesn’t appear to be any discussion of whole building life cycle assessment.

Major Points

The abstract needs to be rewritten, with an emphasis on providing a succinct high-level summary of the study’s methods and a summary of the results. The study specific notation used to refer to the LEED strategies should be removed from the abstract in favor of just writing out the high-level information that is needed to summarize the methods. Ideally some quantitative data from the results should be highlighted in the abstract as well.

A detailed description of the LCA methodology, life cycle inventory, justification for functional unit, and discussion of the system boundary is missing from the manuscript. This may be in error since I note that there is a reference to a Section 3.2.1 (Lines 248-249); however, said section doesn’t appear later on in the manuscript. However, since it appears that Section 3 is intended to be the Results and Discussion, the level of information necessary to properly describe these details would need to appear in the methods – if the study is using the same approach as a previous study then a detailed summary may be sufficient with appropriate citation of the earlier work.

Lines 56 – 59 –“To refer to credits from different systems…” – while I can appreciate ethe challenge in trying to compare the two systems, a table with the mapping needs to be provided with all of the comparisons used in this study.  

Lines 80 – 109 – “In 2022, Pushkar [7] collected forty LEED projects…” – Reading this summary of the article, I’m not sure exactly what the author is trying to communicate regarding the study by Pushkar (2022). Try to distill this down to just the relevant point(s) appropriate for the background to this study.

Lines 110 – 127 – “In 2023, Pushkar [8] analyzed thirty-nine LEED-CI v4 projects…” – ditto above points regarding the level of detail in reference to the cited article, and the notation being used in explaining the study.

Sections 2.1.1 and 2.1.2 – While I understand why these are presented separately, with a bit of revision they can be merged into a single succinct paragraph. Also, I’m not sure why EAmedium is being mentioned in the manuscript if this group wasn’t actually being used in the analysis? Finally, the manuscript needs to explain what the distinction is between the low and high achievement groups.  

Section 2.1.3 – This section needs to be rewritten for clarity. Some of the subsections (e.g., 2.1.3.2, 2.1.3.4) seem to be irrelevant to the study and can either be deleted in full, or quickly addressed (e.g., “sample size not sufficient to evaluate…”).

Section 3.3 – I’m having a really hard time following the justifications for the results in this section given that some of the comparison scenarios don’t seem like a typical application of an LCA methodology. However, that may be due to the missing Section 3.2.1.

Minor Points

The p-value of 0.0000001 should be written out as < 0.0001 given that the other p-values are only be reported to that level of precision.

Author Response

Reviewer 1

In this manuscript the author appears to be attempting to contrast the merits of two different LEED certification strategies. While a critical evaluation of certification strategies is an important part of ensuring that they are meeting objectives of the program conferring the certification, the manuscript fails to establish validity of the methodological approach being used to perform the evaluation. Critically, I note that it appears that an entire section of the manuscript (Section 3.2.1) may have been inadvertently left out of the manuscript and this might be the cause of the lack of clarity. In addition to ensuring the manuscript is complete, the introduction could also use more detail to explain how the author is applying LCA in the LEED context since there doesn’t appear to be any discussion of whole building life cycle assessment.

Major Points

The abstract needs to be rewritten, with an emphasis on providing a succinct high-level summary of the study’s methods and a summary of the results. The study specific notation used to refer to the LEED strategies should be removed from the abstract in favor of just writing out the high-level information that is needed to summarize the methods. Ideally some quantitative data from the results should be highlighted in the abstract as well.

Author’s answer:

The abstract was rewritten according to the reviewer’s comments (please see lines 6-20).

A detailed description of the LCA methodology, life cycle inventory, justification for functional unit, and discussion of the system boundary is missing from the manuscript. This may be in error since I note that there is a reference to a Section 3.2.1 (Lines 248-249); however, said section doesn’t appear later on in the manuscript. However, since it appears that Section 3 is intended to be the Results and Discussion, the level of information necessary to properly describe these details would need to appear in the methods – if the study is using the same approach as a previous study then a detailed summary may be sufficient with appropriate citation of the earlier work.

Author’s answer:

The LCA methodology including functional unit, system boundary, and life cycle inventory was added to the method section (please see lines 103-115, 173-213).

Lines 56 – 59 –“To refer to credits from different systems…” – while I can appreciate ethe challenge in trying to compare the two systems, a table with the mapping needs to be provided with all of the comparisons used in this study.

Author’s answer:

The sentence: “ To refer to credits from different categories…” was just about abbreviation of EAc5 and MRc2. It was deleted. Instead, as per the comments of another reviewer, a table has been added with all the abbreviations used in this study. (please see lines 43-44). 

Lines 80 – 109 – “In 2022, Pushkar [7] collected forty LEED projects…” – Reading this summary of the article, I’m not sure exactly what the author is trying to communicate regarding the study by Pushkar (2022). Try to distill this down to just the relevant point(s) appropriate for the background to this study.

Lines 110 – 127 – “In 2023, Pushkar [8] analyzed thirty-nine LEED-CI v4 projects…” – ditto above points regarding the level of detail in reference to the cited article, and the notation being used in explaining the study.

Author’s answer:

The summaries of both [7] and [8] studies were rewritten to outline the relevant points only (please see lines 78-91).

Sections 2.1.1 and 2.1.2 – While I understand why these are presented separately, with a bit of revision they can be merged into a single succinct paragraph.

Author’s answer:

The sections 2.1.1 and 2.1.2 are merged into a single succinct paragraph (please see lines 118-128).

Also, I’m not sure why EAmedium is being mentioned in the manuscript if this group wasn’t actually being used in the analysis?

Author’s answer:

The EAMedium group has been removed from the text of the manuscript.

Finally, the manuscript needs to explain what the distinction is between the low and high achievement groups. 

Author’s answer:

The lowest performing group on EAC6 credit showed a range of 0 to 8 points, while the highest achieving group on EAC6 credit showed a range of 10 to 24 points (Tables 8 and 9, respectively). The maximum value of this EAc6 credit is 25 points, which is 42% of the minimum value for the LEED Gold certification level. Thus, low or high performance in this EAc6 credit can significantly influence the choice of a gold certification strategy.

Section 2.1.3 – This section needs to be rewritten for clarity. Some of the subsections (e.g., 2.1.3.2, 2.1.3.4) seem to be irrelevant to the study and can either be deleted in full, or quickly addressed (e.g., “sample size not sufficient to evaluate…”).

Author’s answer:

Both the “2.1.3.2 Variability Analysis” and “2.1.3.4 Effect Size Bounds for Green Building Data” subsections were removed from the text of the manuscript.

Section 2.1.3 was reordered. As a result, the section 2.1 contains 2.1.1 “Design of the study”, 2.1.2 “Data analysis”, and 2.1.3 “p-value interpretation”. The section 2.1.2 contains 2.1.2.1 “Statistical analysis” and 2.1.2.2 “Effect size procedure” (please see lines 130-160).

Section 3.3 – I’m having a really hard time following the justifications for the results in this section given that some of the comparison scenarios don’t seem like a typical application of an LCA methodology. However, that may be due to the missing Section 3.2.1.

 Author’s answer:

The section 3.3.1 was clarified (please see lines 307-345) with the additional information and calculation procedure for OE stage-related kWh and P stage-related material quantities were presented in Table 10.

Minor Points

The p-value of 0.0000001 should be written out as < 0.0001 given that the other p-values are only be reported to that level of precision.

 Author’s answer:

When p < 0.0001, the p value was reported as p < 0.0001.

 English was edited (please see the attached certificate).

Reviewer 2 Report

the paper is very well organized, the below issues should be addressed

1-please prepare an abbreviation table in the introduction which helps the reader to follow the manuscript

2-the limitation of the work must be added

3-i want to see the results of various LCA measurements in one single table, it can be provided as a supplementary file as well

4-recommendation for future work is missing

Author Response

Reviewer 2

the paper is very well organized, the below issues should be addressed

1-please prepare an abbreviation table in the introduction which helps the reader to follow the manuscript

2-the limitation of the work must be added

Author’s answer:

The abbreviation table was inserted into Introduction (please see Table 1, lines 43, 44).

3-i want to see the results of various LCA measurements in one single table, it can be provided as a supplementary file as well

Author’s answer:

LCA measurements used in this study were presented in Table 2 (please see lines 202-207) and Table A1 (please see lines 482-486).

4-recommendation for future work is missing

Author’s answer:

Recommendation for future work were added in 6. Recommendation chapter (please see lines 463-468).

 English was edited (please see the attached certificate).

Reviewer 3 Report

The article presents an interesting point of view in the broader discussion about LEED certificates Vs Life Cycle Assesment. An author compared the benefits and limitations of using both methods to describe the energy efficiency or environmental friendship of the investments. The structure of the manuscript is well organized. Despite the multi-use of names of the parameters, an author did best to describe and discuss the results of the study. The paper presents use of LEED-CI v4 in the case study in Shanghai. How about other localisation? Other climate? Other types of constructions? I miss more critical conclusions about LEED-CI v4 in different locations. 

Author Response

Reviewer 3

The article presents an interesting point of view in the broader discussion about LEED certificates Vs Life Cycle Assesment. An author compared the benefits and limitations of using both methods to describe the energy efficiency or environmental friendship of the investments. The structure of the manuscript is well organized. Despite the multi-use of names of the parameters, an author did best to describe and discuss the results of the study. The paper presents use of LEED-CI v4 in the case study in Shanghai. How about other localisation? Other climate? Other types of constructions? I miss more critical conclusions about LEED-CI v4 in different locations. 

Author’s answer:

The influence of other climates and other types of construction was outlined in 5. Limitation section (please see lines 455-461).

Round 2

Reviewer 1 Report

This revision of the manuscript addresses many of the concerns in the original version; however, some concerns still remain. Specifically, it doesn’t appear that the energy mix used in the LCA is appropriate for Shanghai, and it’s still unclear how the author determining the window area per 1 m2 of building area. The former should be fairly straightforward to address since the inventory of power stations in Shanghai can be used; however, the latter might require sensitivity analysis since the precise area occupied on a regular basis can be difficult to access without detailed information concerning interior use.

Lines 112 – 114 – Since the study is focused on buildings in Shanghai, one of the energy mix scenarios should be the current energy mix. Based upon the current list of power stations in Shanghai - https://en.wikipedia.org/wiki/List_of_major_power_stations_in_Shanghai - I would expect natural gas to be included in the mix.

Line 187 – 188 – This needs to be phrased better since the exact wording used in LEED ID+C for Quality Views is that there is a “a direct line of sight to the outdoors via vision glazing for 75% of all regularly occupied floor area.” This is an important distinction as well since is requires evaluation of floor plans and use of space, which can be a challenge to achieve in a study (ex., does a break-room count as regularly occupied floor area or not?). Based upon the manuscript, it’s unclear how the author quantified the floor area in order to be able to use the EQc8 as a metric for the LCA. I note that the author tried to do this quantitatively on lines 324 to 331, but I’m not entirely convinced by the approach the author took since the study cited (Pushkar et al. 2022) appears to primarily focused on the building envelope, therefore interior floor area can be a useful proxy. However, we can reasonably expect that some of the interior floor space will be dedicated to utility purposes and not occupied all of the time and how this accounting is done would impact the results.

Author Response

Dear reviewer,

Please see the attached file with the author's answers.

Round 3

Reviewer 1 Report

While the author is continuing to iterate upon the manuscript, the original points raised in the previous reviews haven’t been sufficiently addressed. To summarize the key issue that I see with the manuscript:

1. The energy mix used by the author still doesn’t appear to be appropriate to the Shanghai context. Additionally, it appears that the 45% coal, 15% oil, 15% natural gas, and 25% photovoltaic mix used is actually based upon the target mix given by Huang et al. (2019). While the use of a national mix may be appropriate, the precise mix should be verified by cross-referencing it with offical documentation. Alternatively, the author could calculate a reasonable local mix for Shanghai based upon the generating stations that are present in the region (https://en.wikipedia.org/wiki/List_of_major_power_stations_in_Shanghai).

2. If I’m understanding the manuscript correctly, the author is assuming that the 60% external wall to floor area applies to all of the projects listed in Tables 8 and 9; however, I’m a bit skeptical of this since it presumes that all of the interiors are comparable in size for this ratio to work. 

3. While the manuscript is largely grammatically correct, the writing is extremely dense and it’s still really hard to follow what’s going on and there are still some typos present in the manuscript (ex., Adidas Shanghai Headquar Office should be Adidas Shanghai Headquarters Office or [sic] should be used if that’s the proper name form the original documentation).

Author Response

Thank you very much for your attention to my article. I apologize for any inaccuracies in this article and I have done my best to improve it. I have responded to your comments and noted the changes in the article. My answers are below.

While the author is continuing to iterate upon the manuscript, the original points raised in the previous reviews haven’t been sufficiently addressed. To summarize the key issue that I see with the manuscript:

  1. The energy mix used by the author still doesn’t appear to be appropriate to the Shanghai context. Additionally, it appears that the 45% coal, 15% oil, 15% natural gas, and 25% photovoltaic mix used is actually based upon the target mix given by Huang et al. (2019). While the use of a national mix may be appropriate, the precise mix should be verified by cross-referencing it with offical documentation. Alternatively, the author could calculate a reasonable local mix for Shanghai based upon the generating stations that are present in the region (https://en.wikipedia.org/wiki/List_of_major_power_stations_in_Shanghai).

Author’s answer:

The local mix for Shanghai based upon the generating stations that are present in the region was used for calculation of the first fuel source scenario (please see lines 17, 114-115, 207-208, 321-322, 377-385, 512-513, 543-544 and Figure 1).

  1. If I’m understanding the manuscript correctly, the author is assuming that the 60% external wall to floor area applies to all of the projects listed in Tables 8 and 9; however, I’m a bit skeptical of this since it presumes that all of the interiors are comparable in size for this ratio to work. 

Author’s answer:

Due to the lack of information about the internal plan of each LEED project, the author considers this study as the first step towards the evaluation of green building in terms of life cycle assessment. The reviewer proposes a more advanced solution to this problem based on actual LEED-CI-certified interior building planes. However, the author will take advantage of the reviewer's suggestion in further research (please see lines 485-489).

  1. While the manuscript is largely grammatically correct, the writing is extremely dense and it’s still really hard to follow what’s going on and there are still some typos present in the manuscript (ex., Adidas Shanghai Headquar Officeshould be Adidas Shanghai Headquarters Officeor [sic] should be used if that’s the proper name form the original documentation).

Author’s answer:

The typos were corrected (please see Tables 8 and 9).

Round 4

Reviewer 1 Report

The author has addressed most of my concerns regarding the methodology of the study, although given the status quo mix of 71.07% coal, 28.08% natural gas, 0.81% wind, and 0.04% photovoltaic, they may wish to consider a mix of wind and photovoltaic for the renewable energy scenario given that Shanghai doesn't seem to have hydropower capability at present. The writing is also still quite dense and hard to follow.

Author Response

The author has addressed most of my concerns regarding the methodology of the study, although given the status quo mix of 71.07% coal, 28.08% natural gas, 0.81% wind, and 0.04% photovoltaic, they may wish to consider a mix of wind and photovoltaic for the renewable energy scenario given that Shanghai doesn't seem to have hydropower capability at present. The writing is also still quite dense and hard to follow.

Author’s answer:

Hydropower was replaced with wind power.

Thus, the mix of wind and photovoltaic for the renewable energy scenario was recalculated (Please see lines 17, 114, 203-204, 316-317, 373-383, 492-493, and Figure 1).